# Research on the dust-control technology of a double-wall attached-ring air curtain on an excavation face

Jingxue Yan[1,2], Baoshan Jia[1,2]*, Xuerong Pan[3], Jinyi Zhang[1,2], Niujun Jia[1,2]

1 College of Safety Science and Engineering, Liaoning Technical University, Fuxing, Liaoning, China, 2 Key Laboratory of Mine Thermal Power Disaster and Prevention, Ministry of Education, Liaoning Technical University, Fuxing, Liaoning, China, 3 China Construction Third Bureau Group Northwest Branch, Xian, Shaanxi, China

* jiabaoshan@lntu.edu.cn

**Data Availability Statement:** All relevant data are within the paper and its Supporting information files.

**Funding:** Natural Science Foundation of Liaoning Province) with grant number (2020-MS-304.

## Abstract

On the basis of the jet theory of airflow fields and the gas–solid two-phase flow theory, we studied the law of dust migration in a simulated dusting space. We used the control variable method and numerical simulation software to explore the airflow field and dust concentration distribution on the working surface of the dusting under different inlet wind speeds and different attached blades of the double-walled annular air curtain. We determined the speed of the inlet of the annular air curtain to be 30 m/s. When the angle of the attached blade was 30˚, the dust concentration of the driver and other workers was controlled below 100 mg/m$^3$, which produced the best dust control effect is the best. Using real data, we built a similar test platform to test the airflow field and dust concentration. Through data measurement and analysis, we proved that a dust control system with a double-wall attached-ring air curtain formed a circulating airflow field that could shield dust and effectively reduce dust concentration in the simulated space. The dust removal efficiency of total dust and exhaled dust reached 98.5% and 97.5%, respectively. We compared the test data and simulation results and concluded that the double-wall attached-ring air curtain could effectively ensure the safety of mine production and provide a better underground working environment for operators.

## 1. Introduction

The integrated excavation face is the focus of efforts to control dust disasters. Dust accumulates easily, and floats and spreads in confined space, introducing significant safety risks to underground work, damaging the physical and mental health of workers, and causing occupational pneumoconiosis and other problems. To solve the problem of dust concentrations that are over the limit, in this study, we examined dust control technology using a double-wall attached-ring air curtain.

Compared with traditional dust control technology, air curtain dust control has good dust control efficiency and offers several advantages [1]. In recent years, many researchers have

**Competing interests:** The authors have declared that no competing interests exist.

conducted numerous studies on wind curtains to reduce mine dust. The earliest application of a wind curtain is noted in the early 20th century by Kemmel [2], who isolated the cold air outside a door by setting a wind curtain on both sides of the door. In the former Soviet Union in the 1950s, Shebelev et al. first applied an air curtain to the working face of a mine roadway [3]. In the 1960s, Grassmuk [4], a French scholar, deduced the lift force of the pressure on both sides of the air curtain and further studied the deduced formula of pressure difference. In the 1970s, Eropob and Megebgeb [5], scholars from the former Soviet Union, based on previous studies, conducted a similar simulation test. Guyonnaud et al. [6] believed that the air curtain inlet area and inlet speed would affect dust control efficiency of the air curtain. In the 1980s, Zhuyun X [7] and others at Northeastern University were able to reduce energy consumption. In 1990, Professor Wang Haiqiao [8] of the Hunan University of Science and Technology conducted relevant research and applied an air curtain to underground dust isolation for the first time. Haiqiao introduced a correlation formula of the concentration ratio between the controlled area and polluted area. In 2000, Yajun L [9] from Liaoning Technical University studied the short-circuit flow field theory and successfully developed a new type of air curtain machine using the superposition principle of three flow fields. In 2008, Chen Caiyun [10] from Liaoning Technical University, carried out a theoretical study on the dust generation mechanism and wind speed attenuation of the excavation face. In 2011, Nie Wen and Cheng Weimin et al. [11] studied and analyzed the spatial distribution law of dust concentration on a downhole integrated excavation surface through mathematical modeling and FLUENT software. Compared with the simulation, the dust concentration was well controlled. In 2012, Liu Zengping and Sun Jingkai [12] studied the problem of excessive dust concentration in the mechanized integrated excavation face of rock. In 2013, Yang Jing, Li Yucheng [13], and others developed dust collection equipment and designed a long pressure and short pumping dust control mode to effectively separate the dust at the header from the staff. In 2015, Daming Zhang and Yundong Ma [14] studied the velocity attenuation law between wind speed at the exit of air curtain in roadway through numerical simulation. In 2020, Liu Ronghua and Zhu Biyong [15] designed a dual-radial swirl-shield ventilation system to address the shortcomings of wall-attached swirl flow.

Scholars have not yet developed the application of air curtain technology and how it may be combined with mine dust treatment. In this study, through theoretical analysis, numerical simulation, and experimental measurement, we explored dust control technology and designed a double-wall attached-ring air curtain system, which could effectively suppress the dust concentration of the excavation face. Based on optimal exit velocity and blade angle of a double-wall attached-ring air curtain determined by numerical simulation, we built a similar simulation test platform for dust control using a double-wall attached-ring air curtain to verify its effect. We effectively reduced the dust concentration of the working face of the shaft excavation to provide a better working environment for underground operators.

## 2. Analysis of dust-control effect of conventional risk on excavation face

### 2.1 Parameter setting and grid division of integrated excavation roadway model

Through the study of dust control on the working face of integrated excavation, the ventilation method of long pressure and short pumping has generally been adopted to control dust. We used FLUENT simulation software to build the physical model of the boring machine and the dust control test platform. Considering the calculation's accuracy and accuracy, the model was reasonably simplified. The section size of the tunneling roadway was set as follows:

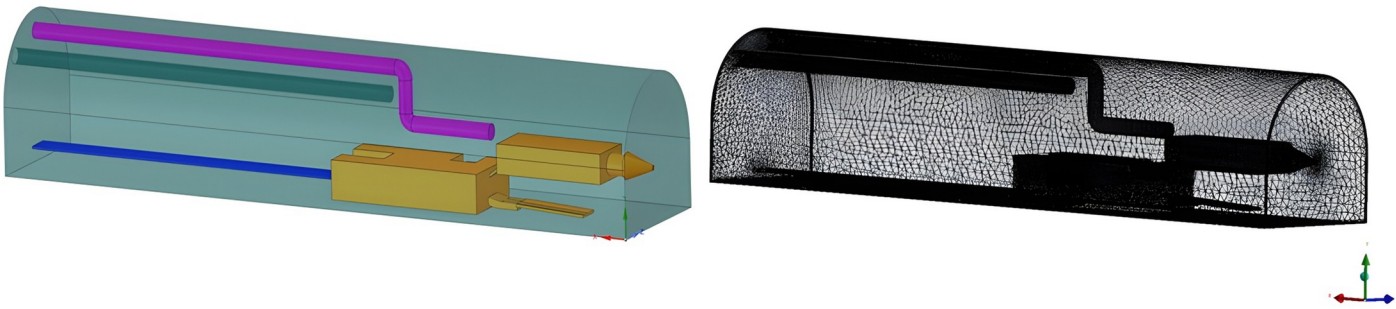

**Fig 1. Physical model and grid division of roadway.**

length × width × height = 30 m × 5 m × 6 m arch roadway. The tunneling machine would draw according to the actual size, with a total length of 14.3 m and a height of 2.5 m. The exhaust duct was 22 m long, the pressure duct was 17 m long, and the staff was at X = 9.5, Y = 1.5, Z = 4. Using the physical field control network, we constructed the model shown in Fig 1.

We used SpaceClaim Direct Modeler (SCDM) to establish the geometric model, imported mesh for mesh division, and imported FLUENT after mesh division. The solver and various parameter settings are shown in Table 1.

## 2.2 Analysis of wind field results

The air was transported to the excavation face through the air pipe, and the air was extracted from the exhaust duct. In the airflow field formed by the excavation face, the wind-wrapped dust was extracted from the excavation space.

**Table 1. Standard table of simulation calculation.**

| Item | Name | Setting Situation |
|---|---|---|
| Solver setup | Solver | Pressure based |
| | | Steady state |
| | | Absolute velocity |
| | | Gravity 9.81 m/s$^2$ |
| Air parameter | Density | 1.225 kg/m$^3$ |
| | Viscosity | 1.7894e-05 kg/m•s |
| | Temperature | 288.16 K |
| Flow field model | Viscous | k-epsilon RNG |
| Boundary condition | Velocity magnitude of inlet duct | 15 m/s |
| | Velocity magnitude of outlet duct | −10 m/s |
| Dust source parameter | Injection type | Surface |
| | Release from | Injection |
| | Material | Coking coal |
| | Maximum diameter | 0.0001 |
| | Mean diameter | 1e-05 |
| | Minimum diameter | 1e-06 |
| | Number of diameters | 50 |
| Wall setting | Bottom plate | Trap |
| | The rest | Reflect |

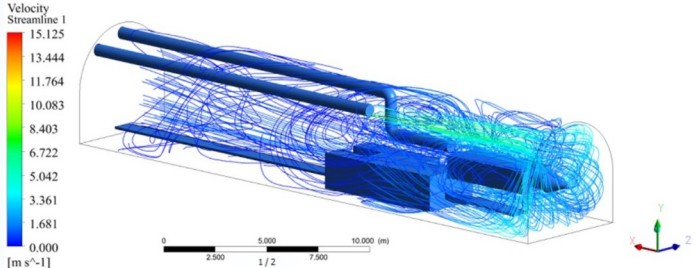

**Fig 2. Trajectory diagram of airflow movement in simulated roadway.**

**(1) Analysis of airflow field in roadway.** As shown in Fig 2, the clean airflow produced by the pressure duct was blocked by the front wall and diffused around the wall. Under the obstruction of the wall surface, a positive pressure area was gradually formed. At this time, under the action of the exhaust duct, a negative pressure area was formed in front of the exhaust duct, and whirlpool flow occurred from the positive pressure area to the negative pressure area. Part of the dust bearing air was sucked in by the exhaust duct, whereas the other part was freely dispersed along the roadway. This result showed that the problem of dust dispersion still existed. To avoid the chance of a simulation experiment, we calculated several groups of airflow trajectories, and changed the positions of pressure duct and exhaust duct on the X axis.

**(2) Wind speed analysis at different sections in the roadway.** As shown in Fig 3, when the airflow was ejected from the pressure air pipe and flowed along the roadway wall, at the roadway section X = 1 m, the pressure air pipe was located above the roadway side, and the airflow from the pressure air pipe was intercepted by the roadway head wall and backflow occurred, resulting in a relatively obvious vortex at the roadway X = 1 m. At X = 6 m, the airflow was affected by the negative pressure of the TBM and the extractor, and there was a vortex around the TBM. At X = 9.5 m, the wind speed at the pressure duct was the largest, reaching 11.76 m/s, and two small whirlpool flows appeared below the pressure duct. The velocity of airflow at X = 9.5–15 m was obviously smaller than that at X = 1–6 m. Because of the joint operation of pressure and exhaust ducts, the velocity of airflow in pressure ducts was greater than that of exhaust ducts, and the cyclone that formed prevented the diffusion of airflow to the entire roadway section. The wind speed near the wall of the pressure duct was very high, higher than the wind speed at other positions, and the flow field was relatively chaotic.

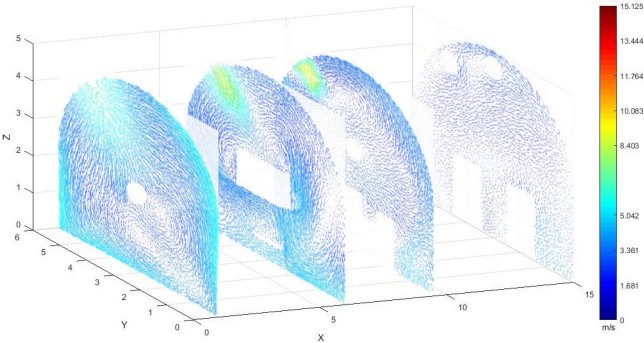

**Fig 3. Wind speed at different X-axis sections in the roadway.**

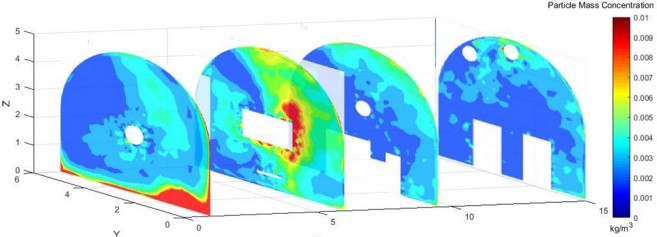

**Fig 4. Cloud map of dust concentration distribution at different X-axis sections.**

## 2.3 Dust concentration analysis

As shown in Fig 4, dust concentration distribution cloud map at different sections in the X-axis direction of the roadway. Under the action of pressure and exhaust duct, some of the dust was prevented from diffusing in the roadway. When the TBM was working, most of the dust was blocked by the vortex airflow. Some of the dust, however, was dispersed in the roadway, the dust concentration was still high at X = 9.5–15 m, and the dust control effect was not ideal.

To more directly observe the distribution of dust concentration in the roadway, we captured a dust distribution cloud map in the Y direction at the normal working height of underground workers.

As shown in Fig 5, Y = 1.5 m and Y = 1.7 m are the main breathing heights of the workers. The cross sections clearly show that the dust concentration was extremely high at the head of the excavation surface, because the dust blew with the wind to the head of the tunnel boring machine and dissipated the return energy. Under the combined action of pressure and exhaust duct, the airflow carried the dust particles along the roadway wall to the head-on section, which was intercepted by the head-on wall and was affected by the negative pressure of exhaust duct. Then, the dust flowed to the opposite side of the pressure duct. The dust was concentrated mainly in front of the staff and near the exhaust duct, and some particles escaped to the rear.

## 3. Double-wall attached-ring air curtain dust control scheme

The simulation analysis showed that the traditional long-pressure and short-suction ventilation method for dust control could not meet the ideal environmental requirements. To better reflect the real situation of airflow and the dust flow field in an integrated excavation space and to investigate the actual production situation, we designed a dust control scheme with a double-wall attached-ring air curtain with reference to the cantilever TBM. We placed the

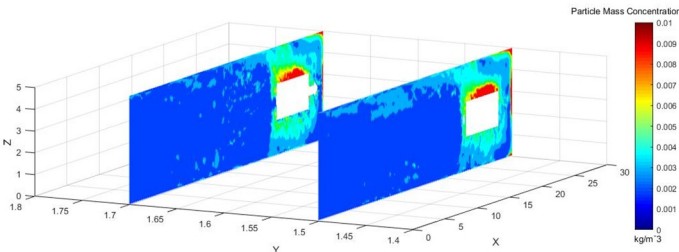

**Fig 5. Staff Y-section dust concentration distribution cloud image.**

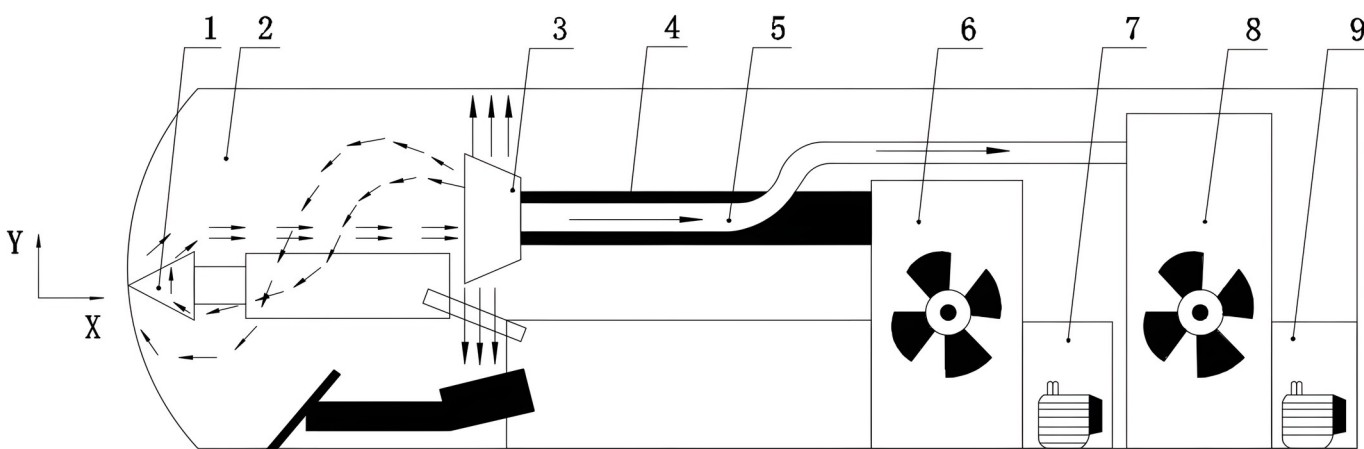

**Fig 6. Dust control scheme schematic.** 1—Excavator, 2—Excavator roadway, 3—Ring air curtain, 4—Pressure air duct, 5—Extraction air duct, 6—Pressure fan, 7—Pressure fan motor, 8—Extraction fan, 9—Extractor electrical machine.

attached-ring air curtain machine behind the head of the excavator and connected the air inlet pipe interface of the ring air curtain machine with the pressure fan. We installed the exhaust pipe in the center of the ring air curtain and connected the exhaust pipe to the exhaust fan, as shown in Fig 6.

The arrow in Fig 6 indicates the airflow path and direction. The compressed fan delivered gas to the ring air curtain machine through the pressure duct, and the exhaust duct was installed in the center. Under this joint action, the airflow field shown by the arrow was formed. A stable airflow field can be formed gradually through the ring air curtain, and dust flowing to the central area of the ring air curtain would be extracted, and wind curtain isolation dust would be formed [16].

The structure of annular air curtain machine is shown in Fig 7. Annular air curtain was provided with inner (outer) ring nozzles. Each outer-ring nozzle was equipped with wall attachment blades to form wall attachment effect [17]. We also analyzed the principle of a wall attachment effect [4] and a low-pressure area appeared on the wall of the attached side, so that the pressure on the side was $P_1$, the other side was $P_2$, the centripetal force was $P_1$-$P_2$ when the airflow passed through the wall attachment, and the centrifugal force was generated by inertia. When the jet [18–20] was stable, the two forces should be equal, as follows:

$$\frac{J_0}{R} = P_1 - P_2,\tag{1}$$

where $J_0$ represents the inertia of the jet; and $R$ represents the curved radius of the jet.

The expression of jet inertia is as follows:

$$J_0 = \rho V_0^3 w,\tag{2}$$

where $\rho$ represents fluid density, kg/m³; $V_0$ represents the velocity of the jet at the nozzle, m/s; and $w$ represents the nozzle width, m.

Through this analysis, we concluded that the factors influencing the wall attachment effect included the wall attachment curvature, jet velocity, and nozzle width. Therefore, to explore the influence of the double-wall attached annular air curtain on dust control of the excavation

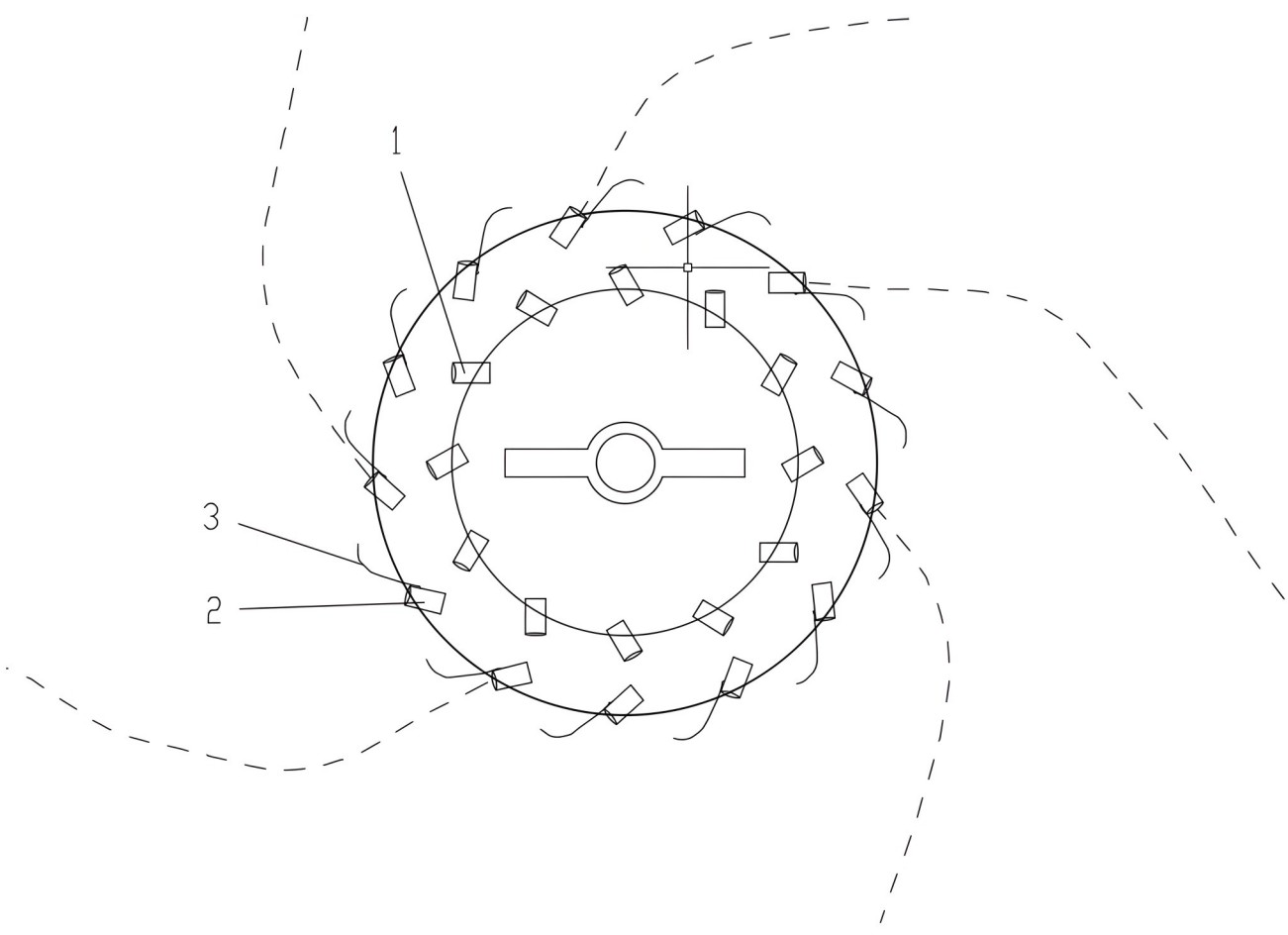

**Fig 7. Structure of ring air curtain machine.** 1—Inner-ring nozzle, 2—Outer-ring nozzle, 3—Wall blade.

face, we set the initial velocity of different air curtain inlets and the angle of the wall-attached blade for a simulation designed to achieve the best dust control effect.

## 4. Numerical simulation of the dust-control effect of a double-wall attached-ring air curtain on the excavation face

### 4.1 Parameter setting and grid division of an integrated excavation roadway model

The inner diameter of the ring was 1.6 m, the outer diameter was 1.8 m, the inner ring was arranged with 10 nozzles, and the outer nozzles and the wall attachment were arranged with 20 nozzles. We constructed this model by using physical field control network, as in Fig 8.

The solver and parameter settings are shown in Table 2.

### 4.2 Analysis of simulation results from changing the initial velocity of air curtain inlet inlet

We set three different inlet wind speeds of 25 m/s, 30 m/s, and 35 m/s in the simulation. We used the incompressible k-ε turbulent field model to calculate the airflow field of the driving face and used the fluid flow particle tracking module for the release of dust particles.

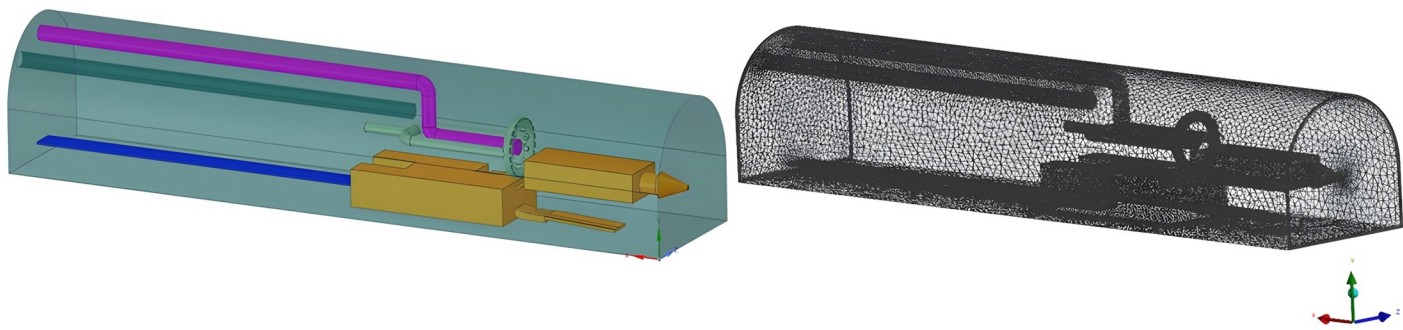

**Fig 8. Physical model and mesh generation of annular wind curtain with double attached walls.**

**Table 2. Standard table of simulation calculation.**

| Item | Name | Setting situation |
|---|---|---|
| Solver setup | Velocity magnitude of inlet duct | 25, 30, 35 m/s |
| | Max number of steps | 30,000 |
| | Maximum diameter | 0.0001 |
| | Mean diameter | 1e-05 |
| | Minimum diameter | 1e-06 |
| | Number of diameters | 50 |
| Wall setting | Bottom plate | Trap (capture) |
| | The rest | Reflect (rebound) |

**4.2.1 Analysis of wind field results.** As shown in Figs 9–11, the migration of the airflow field moved in two directions. The airflow was ejected with the jet cavity, and the air from the outer nozzle formed a "spiral" airflow field that advanced forward. Part of the airflow field was pumped away by the negative pressure of the exhaust duct, and the other part was blown back to the head by the backflow of the roadway to circulate. The wall attachment changed the direction of part of the airflow and pressed toward the roadway wall along with the wall movement. Because of the angle and distance of the air inlet, the airflow moved toward the driving head and was offset as a result of the influence of the negative pressure of the exhaust duct. Finally, a stable circulating eddy current field was formed in front of the air duct.

As shown in Figs 12–14 when the initial velocity of the inlet was different, the airflow migration trajectory at each section was stable and tended to be similar, and the vortex phenomenon of the flow field was obvious because of the suction effect of the airflow field. When the air duct injected fresh airflow into the roadway, the airflow moved forward along the roadway wall to the head section to form a swirling wind. When this part of the airflow moved to the pressure tuyere, it was driven to the working face by the airflow band flowing out of the pressure duct. A vortex backflow area was formed in the area between the compressed air inlet and the roadway head close to the floor. When the inlet speed was 25 m/s and 35 m/s, a large amount of dusty airflow was stacked under the excavator, and the airflow behind the wind was more noticeable, which affected the diffusion and migration of dust in the roadway. In contrast, when the exit speed was 30 m/s, the accumulation of dust air under the excavator was reduced, and the dust air fields in the areas where personnel operated the excavator were sparse and stable.

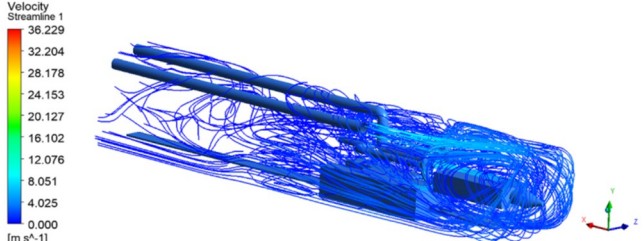

**Fig 9. Airflow track diagram of roadway when exit velocity is 25m/s.**

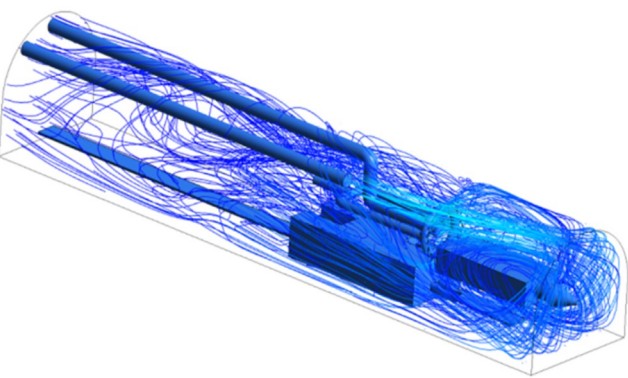

**Fig 10. Airflow track diagram of roadway when exit velocity is 30m/s.**

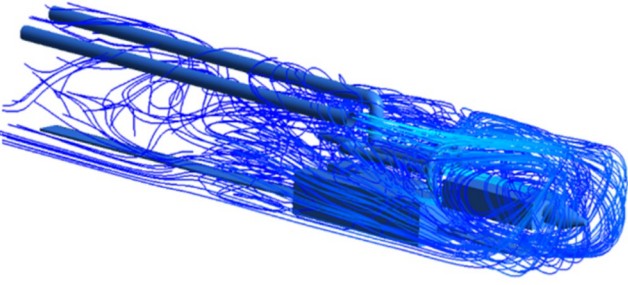

**Fig 11. Airflow track diagram of roadway when exit velocity is 35m/s.**

**4.2.2 Dust concentration analysis.** Fig 10 shows the dust concentration control at different sections at different exit speeds of 25 m/s, 30 m/s, and 35 m/s.

Figs 15–17 shows that the dust particles migrated to the roadway wall with the airflow of the pressure duct. Because of the joint action of the exhaust duct and the air curtain, the dust concentration gradually decreased along the X axis. Most of the dust was pumped away by the suction flow, and part of the dust was shot into the roadway wall by the nozzle. Some of the dust that was not pumped out flowed toward the head with the wind, was intercepted by the wall, and then returned to the direction of the TBM. Because of the obstruction of the fuselage,

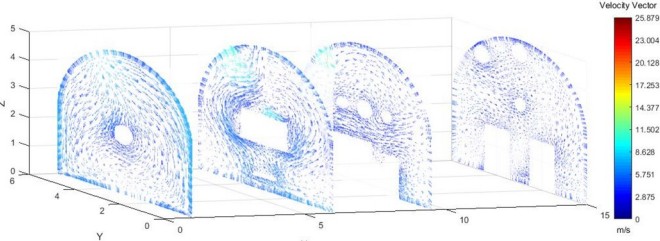

**Fig 12. Wind speed in different X-axis sections of the tunnel when the exit speed is 25m/s.**

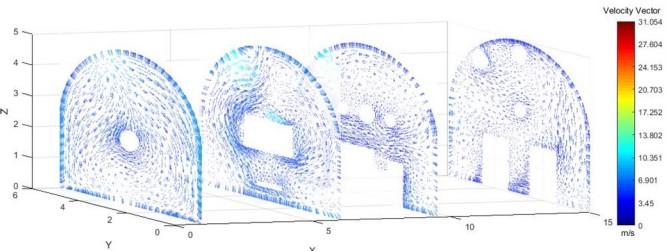

**Fig 13. Wind speed in different X-axis sections of the tunnel when the exit speed is 30m/s.**

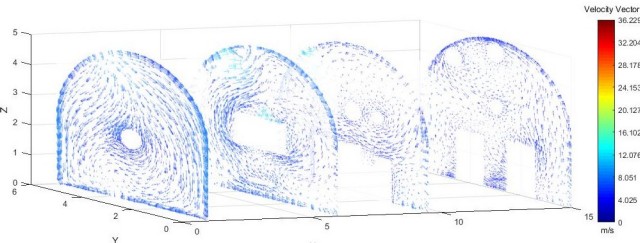

**Fig 14. Wind speed in different X-axis sections of the tunnel when the exit speed is 35m/s.**

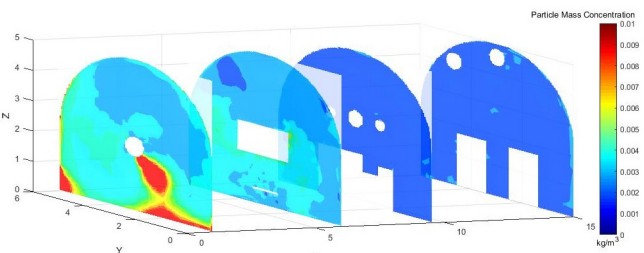

**Fig 15. Dust concentration diagram when the air curtain inlet speed is 25m/s.**

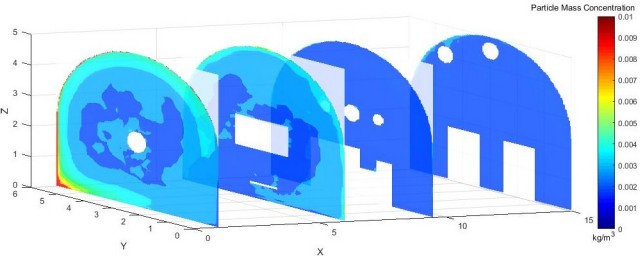

**Fig 16. Dust concentration diagram when the air curtain inlet speed is 30m/s.**

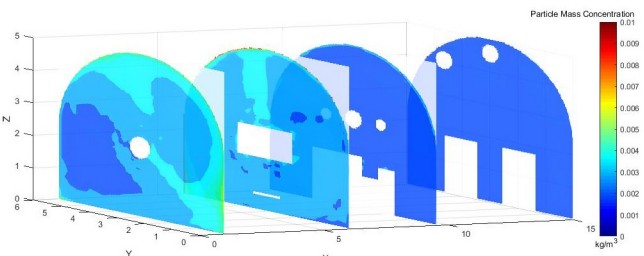

**Fig 17. Dust concentration diagram when the air curtain inlet speed is 35m/s.**

other dust was deposited in the lower part of the TBM to form a stack. The other part was dispersed to the rear of the roadway with the wind flow, but because of the backflow, only a small amount of dust flowed to the rear and diffuses. The dust with large particle size basically settled within 9.5 m from the head, whereas the dust with a small particle size moved farther with the wind flow. Therefore, the size of the dust particle was an important factor affecting the settling speed of dust in the roadway.

Figs 18–20 shows concentration cloud maps at different speeds in the Y direction at the height of the workers.

Fig 21, on the X-Y plane of the roadway, compares the cloud map of dust concentration of the operators at different inlet speeds. According to the in-depth analysis, the dust concentration was higher at the excavation face. Because of the dust control effect of the air curtain and the suction effect of the exhaust duct, most of the dust was removed, and the dust control was

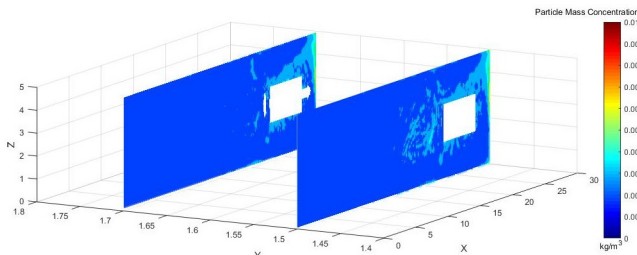

**Fig 18. Dust concentration diagram when the air curtain inlet speed is 25m/s.**

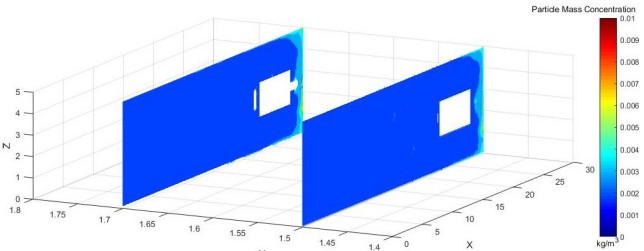

**Fig 19. Dust concentration diagram when the air curtain inlet speed is 25m/s.**

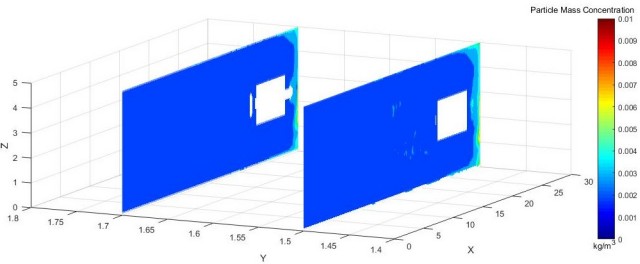

**Fig 20. Dust concentration diagram when the air curtain inlet speed is 25m/s.**

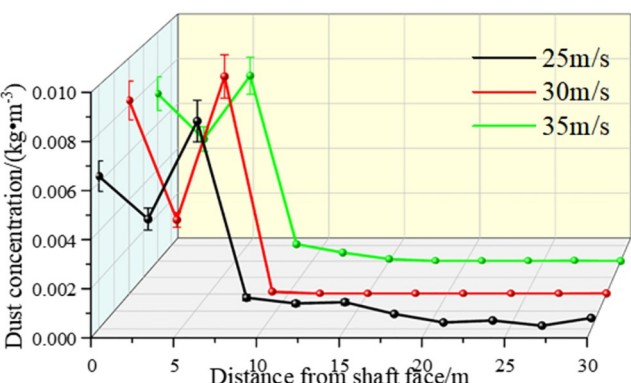

**Fig 21. The dust concentration at the X-axis section was distributed along the path at different exit speeds.**

close to ideal. When the inlet wind speed was 25 m/s, however, the wind speed did not reach the optimal jet speed, which could not prevent the dust from escaping to the driver. When the wind curtain inlet speed was 35 m/s, the high wind speed blew the dust that had accumulated on the ground, and the wind flowed to the rear of the tunnel diffusion, Y = 1.5 m, X = 9.5–30 m, and there was still some dispersed dust. When the exit wind speed was 30 m/s, especially after the X = 9.5 m section, the dust concentration was reasonably controlled.

### 4.3 Analysis of simulation results from changing the angle of the wind curtain attached blade

To study the influence of the change of the angle of the attached blade on the dust isolation and collection system of the air curtain, we changed the angle between the attached blade and the annular air curtain by keeping the size of the air strip slit and the initial velocity of the air inlet unchanged. According to the simulation results, the initial jet velocity V was set to 30 m/s and the angle of the blade attached to the air curtain was 20˚, 30˚, and 45˚. We carried out simulations on the airflow field, dust migration, and dust control of the double-attached air curtain under these three conditions.

**4.3.1 Analysis of wind field results.**   As shown in Figs 22–24, a relatively stable airflow field formed in front of the wind curtain at all three angles. Compared with the airflow field without a wind curtain, we found that the airflow field with a wind curtain had more obvious annular vortices and less miscellaneous and irregular airflow. In addition, the wall adhesion effect generated by the blades effectively improved the airflow field. The dust-bearing air generated by different angles of the wall-attached blades behind the wind varied significantly:

1. Airflow track: When the wall attached blade was 20˚, the airflow track was chaotic, and the dusty airflow was partially returned and partially blown directly to the rear wall; when the blade attached to the wall was 30˚ and 45˚, the airflow path was obvious, and more dusty airflow returned to the front of the ring air curtain, which was conducive to operation.

2. Annular air curtain wall-attachment interface: When the angle of the wall attachment blade was 20˚ and 45˚, no obvious wind curtain wall formed between the air curtain and the roadway wall, resulting in dust airflow in the integrated excavation space escaping to the rear; when the angle of the attached blade was 30˚, the dust airflow in the rear working space was reduced, the airflow flow path and the wall angle increased significantly, and part of the airflow reached nearly 90˚ under the wall attachment effect. At this time, a wind curtain wall was formed.

As shown in Figs 25–27, when the air duct injected fresh airflow into the roadway, the airflow moved forward along the roadway wall to the head-on section to form a swirling wind. The swirling wind appeared between X = 1 m and X = 6 m, indicating that under the action of the wall duct, the swirling wind moved along the perimeter wall of the roadway and also moved in a head-on direction. At X = 6–9.5 m, the airflow gradually became a swirling wind around the wall of the roadway, and the entire roadway section was completely closed. The velocity of X = 9.5–15 m airflow was smaller than that of X = 6–9.5 m. Under the combined

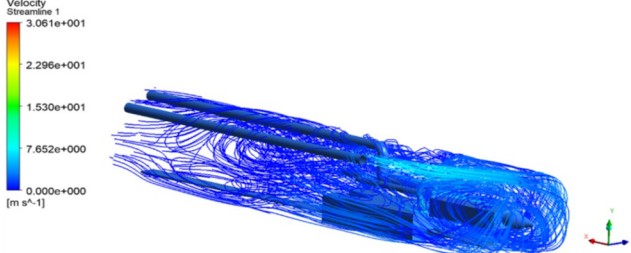

**Fig 22.  The airflow trajectory of the roadway is simulated when the Angle of the attached blade is 20˚.**

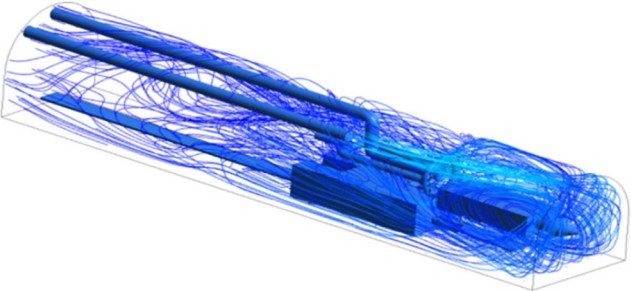

**Fig 23. The airflow trajectory of the roadway is simulated when the Angle of the attached blade is 30˚.**

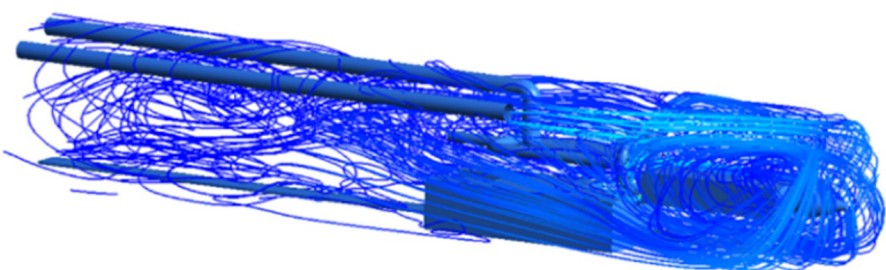

**Fig 24. The airflow trajectory of the roadway is simulated when the Angle of the attached blade is 45˚.**

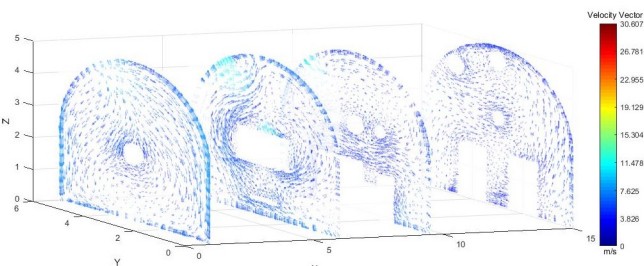

**Fig 25. Wind speed in different X-axis sections of the roadway when the Angle of the attached blade is 20˚.**

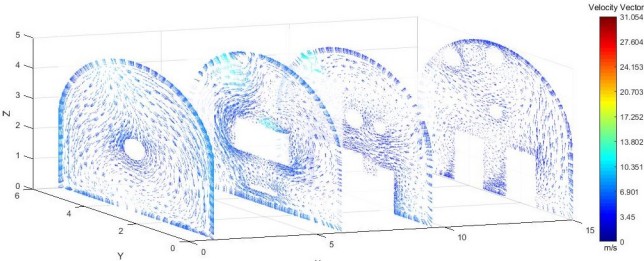

**Fig 26. Wind speed in different X-axis sections of the roadway when the Angle of the attached blade is 30˚.**

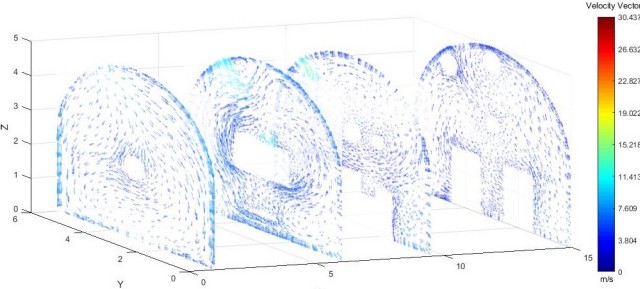

**Fig 27. Wind speed in different X-axis sections of the roadway when the Angle of the attached blade is 45˚.**

action of a double-wall attached annular air curtain and long-pressure and short-suction dust control device, the swirling wind that formed prevented the diffusion of dust to the roadway section. The velocity of airflow on the pressure side was greater than that on the suction side, and the velocity of airflow near the roadway wall was greater than that at the center of the roadway.

**4.3.2 Dust concentration analysis.** Figs 28–30 shows the control of dust concentration at different sections of the roadway when the angle of the attached blade was 20˚, 30˚, and 45˚.

According to the simulation results, the dust concentration at X = 1 m was the highest, reaching $8.15*10^{-3}$ kg/m³. When the angle of the blade attached to the wall was 30˚, the average cross-section concentration was reduced, and the low concentration area was the largest. When the angle of the blade attached to the wall was 20˚, the concentration separation layer was more obvious, and the dust concentration near the wall was higher, which could not be carried away by the airflow. When the angle of the blade attached to the wall was 45˚, the concentration under the TBM was higher and the airflow field was not good. The X = 15 m section reflected the concentration of the rear working space, and the dust concentration in this area affected the health of the staff. According to the Coal Mine Safety Regulations, when the maximum allowable concentration of respirable dust in the working space attached to the wall blade was 45˚, some areas exceeded the prescribed standard. When the blade attached to the wall was 20˚, only a small part exceeded the specified standard. When the blade attached to the wall was 30˚, the prescribed standard was basically met.

Figs 31–33 shows the concentration cloud maps at three speeds in the longitudinal Y direction at the height of the workers:

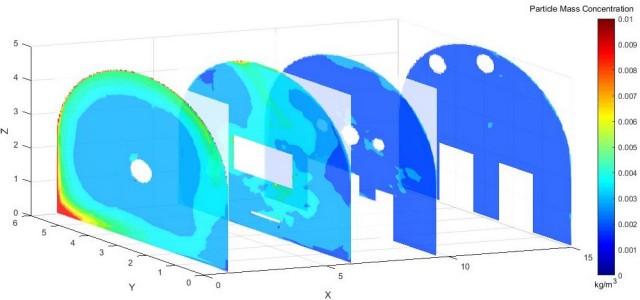

**Fig 28. Cloud image of dust concentration when the Angle of blade attachment is 20˚.**

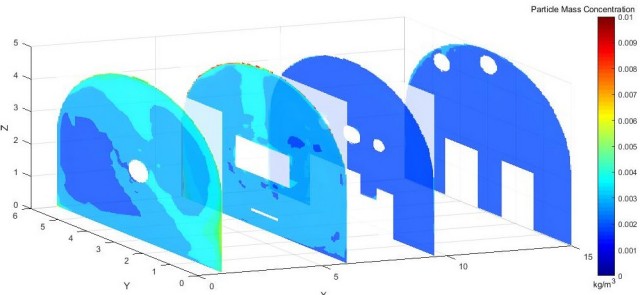

**Fig 29. Cloud image of dust concentration when the Angle of blade attachment is 30˚.**

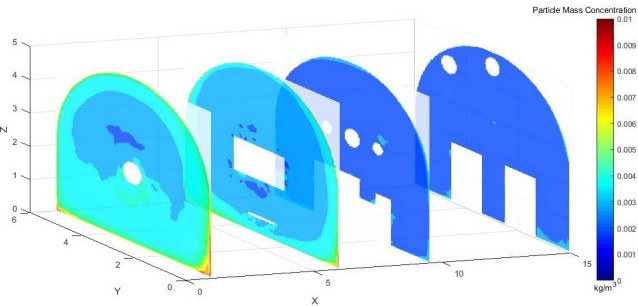

**Fig 30. Cloud image of dust concentration when the Angle of blade attachment is 45˚.**

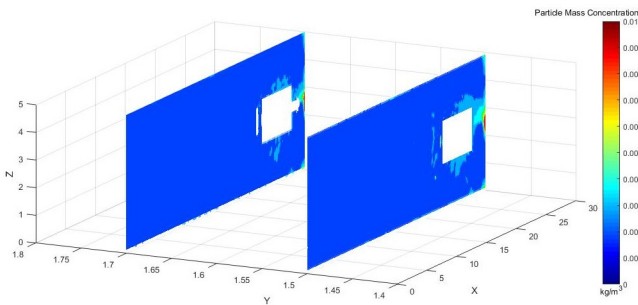

**Fig 31. Cloud image of dust concentration when the Angle of blade attachment is 20˚.**

The dust concentration was higher at the head of the driving machine, and the dust concentration was more obvious when the attached blade was 20˚. When the attached blade was 30˚ and 45˚, however, the concentration in front of the boring head was significantly reduced, which resulted in the formation of faults and less dust escaped. In the rear working space, the dust concentration was higher when the attached blade was 20˚ and 45˚. When the blade attached to the wall was 30˚, there was a low concentration of dust at Y = 1.5 m, a very small amount of dust escaped backward, and the dust concentration reached the lowest at Y = 1.7 m.

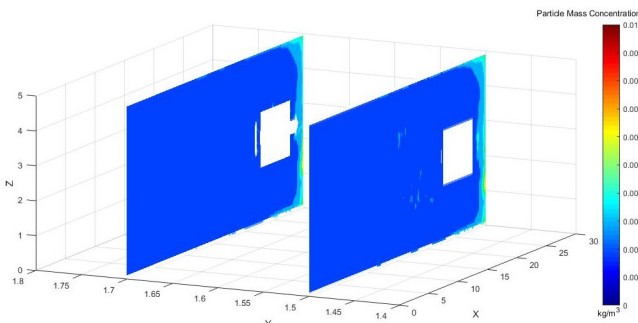

**Fig 32. Cloud image of dust concentration when the Angle of blade attachment is 30˚.**

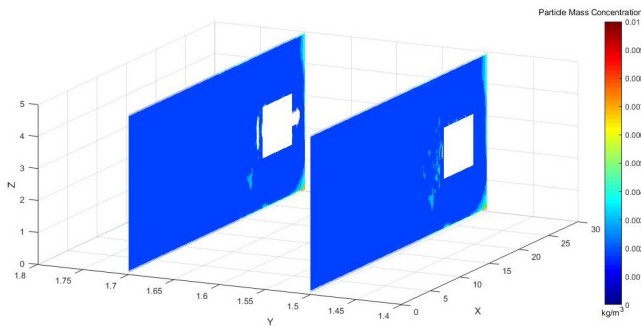

**Fig 33. Cloud image of dust concentration when the Angle of blade attachment is 45˚.**

Through numerical simulations, we collected the dust concentrations at different distances from the excavation face using different angles of wall-attached blades; the changes are shown in Fig 34. Combined with the analysis of airflow field migration results, the following conclusions can be drawn: (1) Under the condition that the air curtain nozzle inlet size was unchanged and the initial velocity of the inlet was unchanged, the different angles of the wall-

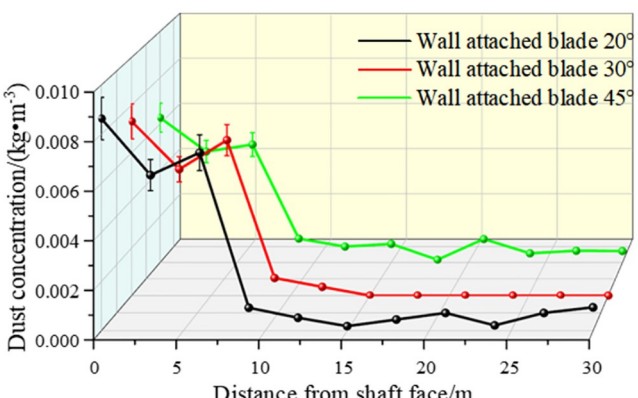

**Fig 34. The dust concentration was distributed along the X-axis section using different blade attachment angles.**

attached blades had an impact on the dust control effect of the excavation face. The simulation results showed that when the wall-attached blade was at 30˚, the dust concentration of each X-axis section was relatively low, and the dust control effect was the best. (2) As shown in Fig 17, the dust concentration was higher when the distance from the excavation face was 0–6 m. At the distance of 6–9.5 m from the excavation surface, the dust concentration dropped sharply. This is the area where the air curtain was located, which proved the effect of a double-wall attached-ring air curtain on dust control. When the distance from the excavation surface was 9.5–30 m, the surface was the active area of the staff. This space had strict requirements for dust. When the exit speed of the air curtain was 30 m/s and the angle of the wall blade was 30˚, the dust concentration in this area was always maintained in the specified range.

## 5. Experimental study of a dust control system with an air curtain on the excavation face

### 5.1 Test platform construction

The test was similarly scaled down according to the real specifications, and the test environment specification was as follows: height 2 m × width 1.8 m × length 5 m. To simulate the real underground integrated excavation face, we installed a dust-generating machine at the front of the test environment instead of the boring machine. We used a YYF-2-6312 fan to provide airflow, set up two pressure air ducts to connect with the air curtain, and installed an air extractor in the center of the ring air curtain.

Fig 35 shows the composition of the test device. The self-developed and designed double-wall attached steam cyclone curtain dust control device was in the shape of a "ring" and was placed behind the head of the excavator. The equipment included a fan, a pressure air duct, an extraction air duct, a nozzle, a dust generator, a ring air curtain, and a wall-attached blade. The self-developed ring air curtain had a 0.6 m outside diameter and a 0.5 m inside diameter. The nozzles were attached to the inner and outer sides of the ring at 30˚, and the outer jet nozzles were attached to the wall for drainage. The blower adopted a centrifugal AC blower with a single-phase capacitor running motor, and the one-way closed roadway effectively restored the real working roadway under the mine. At 1.2 m in front of the ring air curtain, the dust generator simulated the high-concentration dust produced at the coal-cutting operation of the TBM

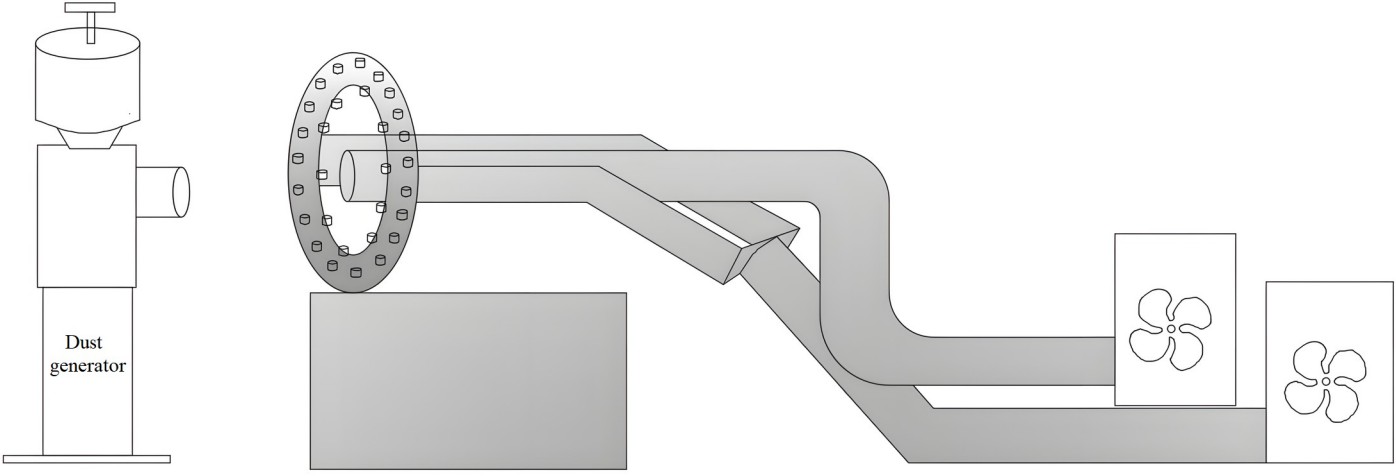

**Fig 35. Test platform model.**

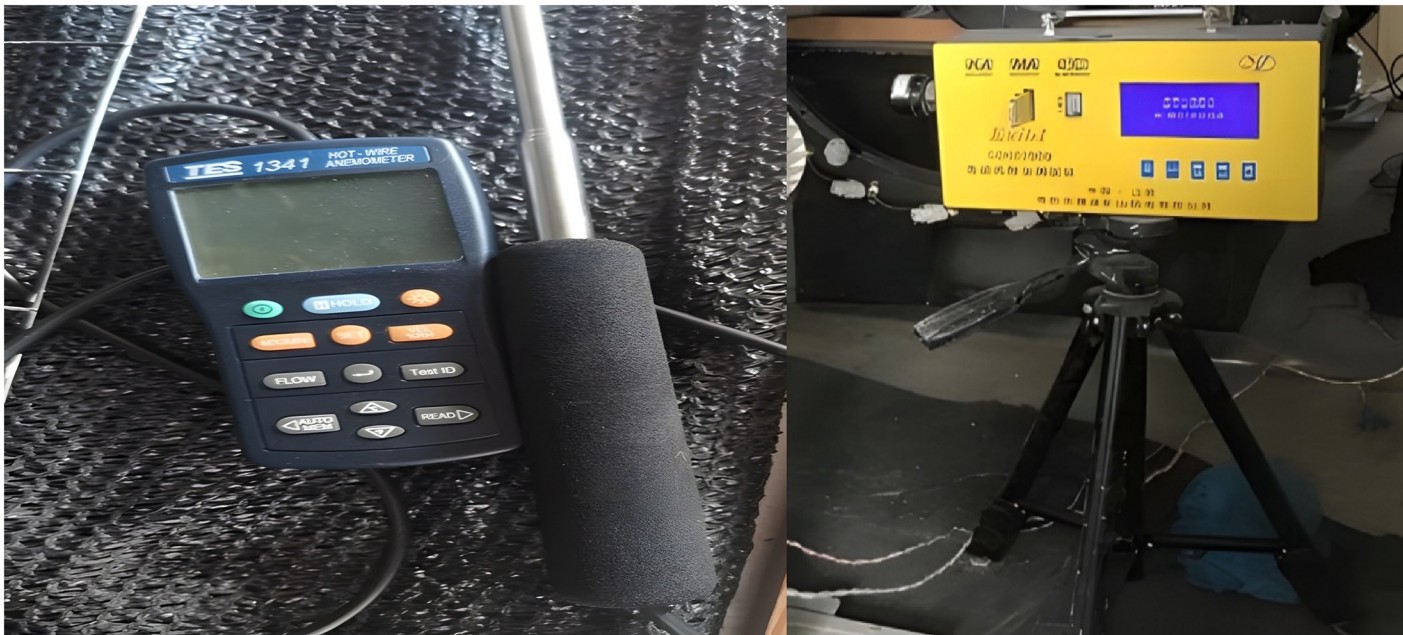

**Fig 36. Hot-wire anemometer and dust measuring instrument.**

cutter. We added talc powder as the test dust to restore the underground working environment.

In terms of equipment connection, connected the compressed air cylinder of the annular air curtain machine with the compressed air blower, and connected the exhaust cylinder with the exhaust fan. The motor of the compressor controlled the fan, and the motor of the exhaust fan controlled the exhaust fan. The two motors do not interfere with each other. The pressure air duct transmitted gas to the ring air curtain, and the exhaust fan pumped air through the exhaust duct in the center of the ring air curtain.

As shown in Fig 36, the hot-wire anemometer was mainly used to measure the wind speed of various parts in small and medium-sized test equipment. The thermal-sensitive wind speed probe was suitable for high-precision wind speed measurement. The telescopic rod can be adjusted arbitrarily. It can not only measure the inlet and inlet wind speeds, but also facilitate movement in the roadway, and accurately measure wind speeds in local complex locations. As shown in Fig 36, the CCHZ1000 direct-reading dust detector used the principle of infrared light absorption to measure the concentration of dust floating in the ambient air. Not only can it accurately measure the concentration of total dust and respirable dust, but also accurately reflected the degree of dust pollution in the workplace, providing a reliable basis for accurately evaluating the hygienic status of the working environment. During the operation process, put the new filter membrane into the filter membrane holder first, cleared the data of the dust tester, set the sampling time, replaced the dust sampling head or the whole dust sampling head, and placed the tester facing the direction of the wind flow. Wait for the sampling time to complete and then read the data.

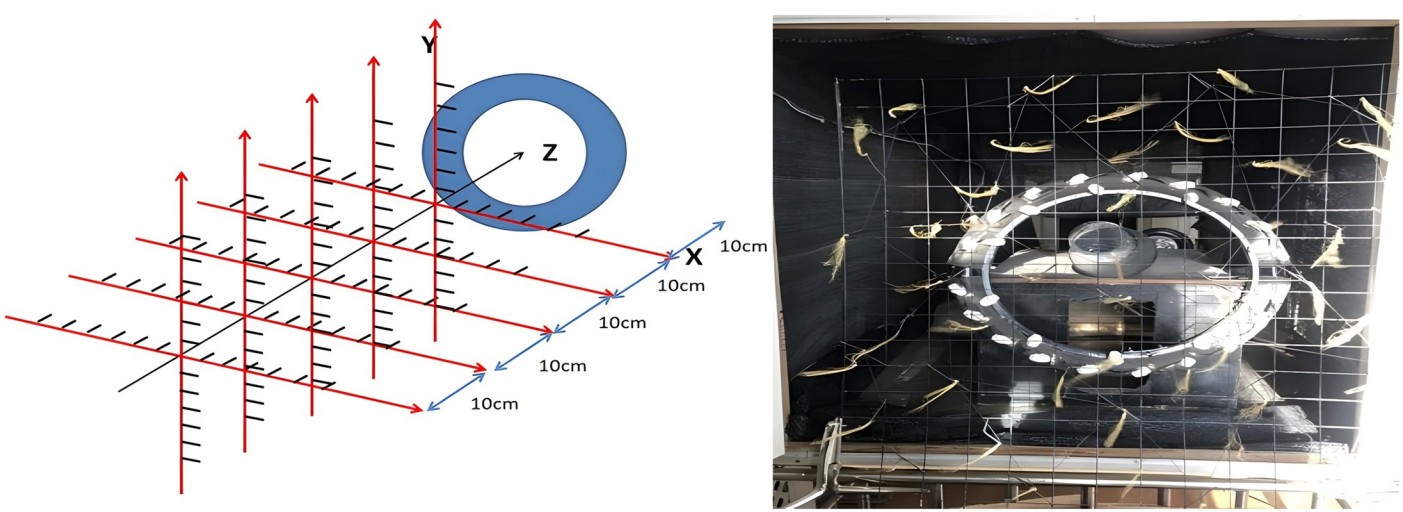

**Fig 37. Layout of wind speed measuring points of air curtain equipment and display of migration track of air field at the test site.**

## 5.2 Test point arrangement and result analysis

**5.2.1 Wind speed measuring point layout and result analysis.** To accurately obtain the airflow field movement law of the dust control equipment and measure the effective dust control wind speed, we selected equidistant points on different planes for data testing.

As shown in Fig 37, we selected a plane at 10 cm, 20 cm, 30 cm, 40 cm, and 50 cm away from the center of the air curtain for the wind measurement point and arranged 12 measuring points along the X and Y axes, with a distance of 7 cm between each measuring point. As shown in Fig 37, the flexible ribbon can be fixed with a metal mesh to accurately mark the location of the wind speed test point and test the direction of the speed change point to facilitate the analysis of airflow trajectory.

As shown in Figs 38–42, along the X and Y axes, the velocity decreased with the distance from the center of the circle, and the attenuation velocity decreased linearly with the increase of the distance. Along the Z axis, the farther away from the dust control equipment, the lower

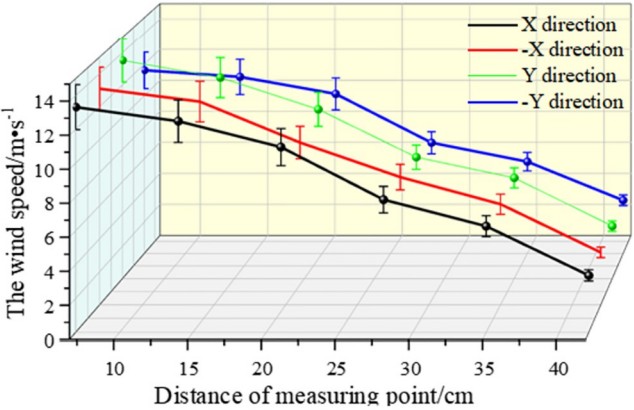

**Fig 38. Plane velocity attenuation at a distance of 10cm from the curtain.**

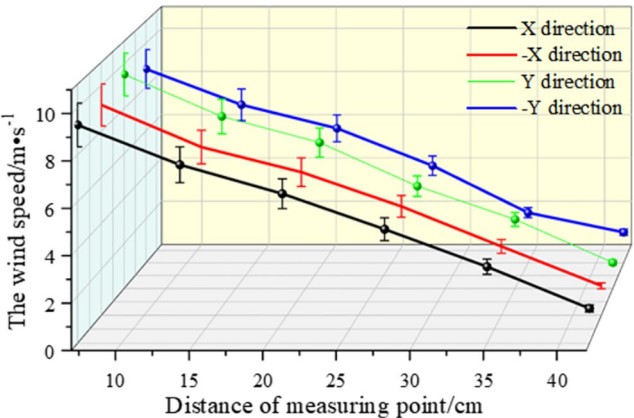

**Fig 39. Plane velocity attenuation at a distance of 20cm from the curtain.**

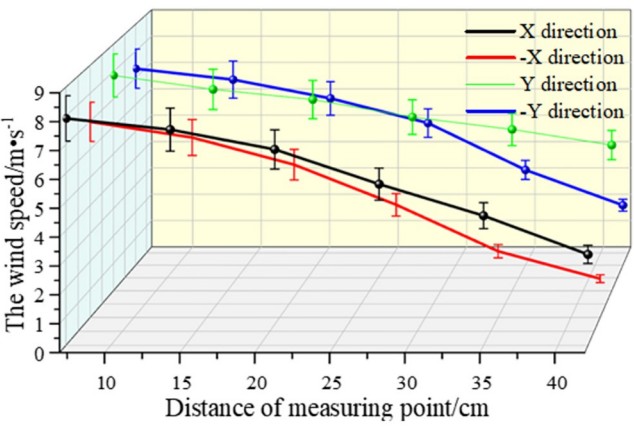

**Fig 40. Plane velocity attenuation at a distance of 30cm from the curtain.**

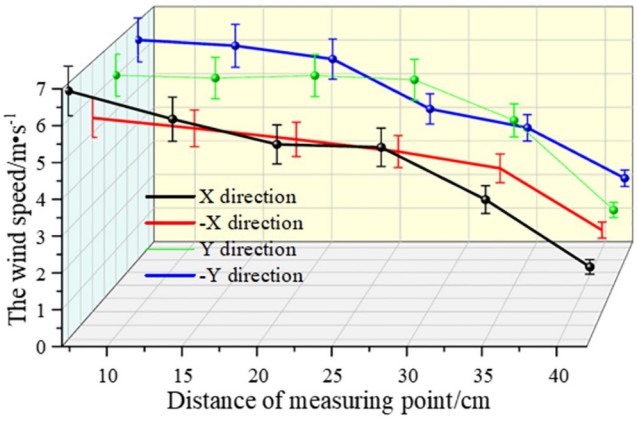

**Fig 41. Plane velocity attenuation at a distance of 40cm from the curtain.**

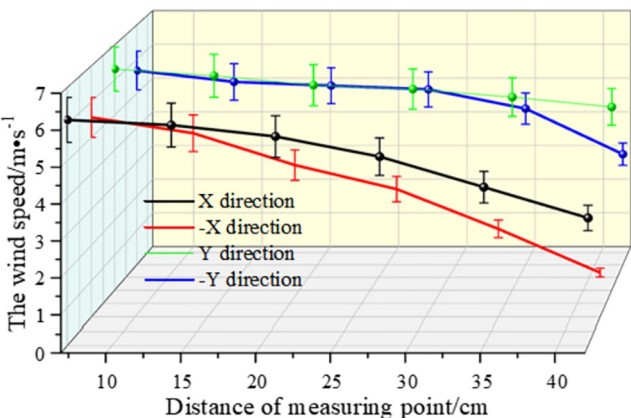

**Fig 42. Plane velocity attenuation at a distance of 50cm from the curtain.**

the wind speed. At 10 cm away from the dust control equipment, the speed was attenuated from 13.31 m/s to 3.17 m/s, and at 20 cm, the speed was attenuated from 9.52 m/s to 1.15 m/s. Because the distance between 0 and 20 cm was closer to the exhaust duct, the wind speed closer to the center was larger and was subjected to the joint action of the wall-attached airflow and the negative pressure of the exhaust duct. At 30 cm away from the dust control equipment, the airflow fluctuated more, the wind speed along the Y axis had less of an influence than the wind speed along the X axis, and the speed attenuation was faster than that at the 20 cm plane. The airflow field at 40–50 cm away from the dust control equipment tended to be stable, the attenuation speed of X and Y axis was small, and the wind speed decreased from 6.31 m/s to 3.15 m/s. Because the side window of the dust generator was placed at the head of the equipment, it had a certain influence on the migration of airflow, so that the X, -X, Y, and -Y axes were somewhat different. Through this analysis, we found that a regular spiral airflow field formed at the front end of the ring air curtain, and the 20 cm and 30 cm planes were the primary dust control areas and were subjected to the action of the suction flow of the exhaust duct to a certain extent.

**5.2.2 Dust concentration measuring point layout and result analysis.** To study the dust control performance of the double-attached air curtain dust control device, we placed a dust generator on the front surface of the test box. We used talc powder as the dust source material in the test and selected four measuring points along the axial section, taking the air curtain as the origin point: (1) (0.5, 1.5); (2) (−0.5, 1.5); (3) (−1, 1.5); (4) (−1.5, 1.5), and (5) (−2.0), (1), (1), (2), (−1.5), (−2.0), (1, 1.5) test objects and test points.

As shown in Fig 43, when the dust control equipment was not started, the dust filled the entire roadway at a very high concentration, and the total dust concentration at the dust source point reached 809.1 mg/m$^3$. The dust spread from the head of the excavation face along the roadway as a whole, and the dust concentration reached 262.3 mg/m$^3$ at the position of 200 cm. Dust failed to decrease through natural settlement, far beyond the national standards, which seriously affected the normal production conditions for underground personnel. After the dust control device was opened, the concentration of total dust and respirable dust was significantly reduced by the dust control action of the steam cyclone curtain field. The total dust removal efficiency reached 97%, and the exhaled dust removal efficiency reached 95%. The double-wall attached steam cyclone curtain dust control test platform effectively controlled the

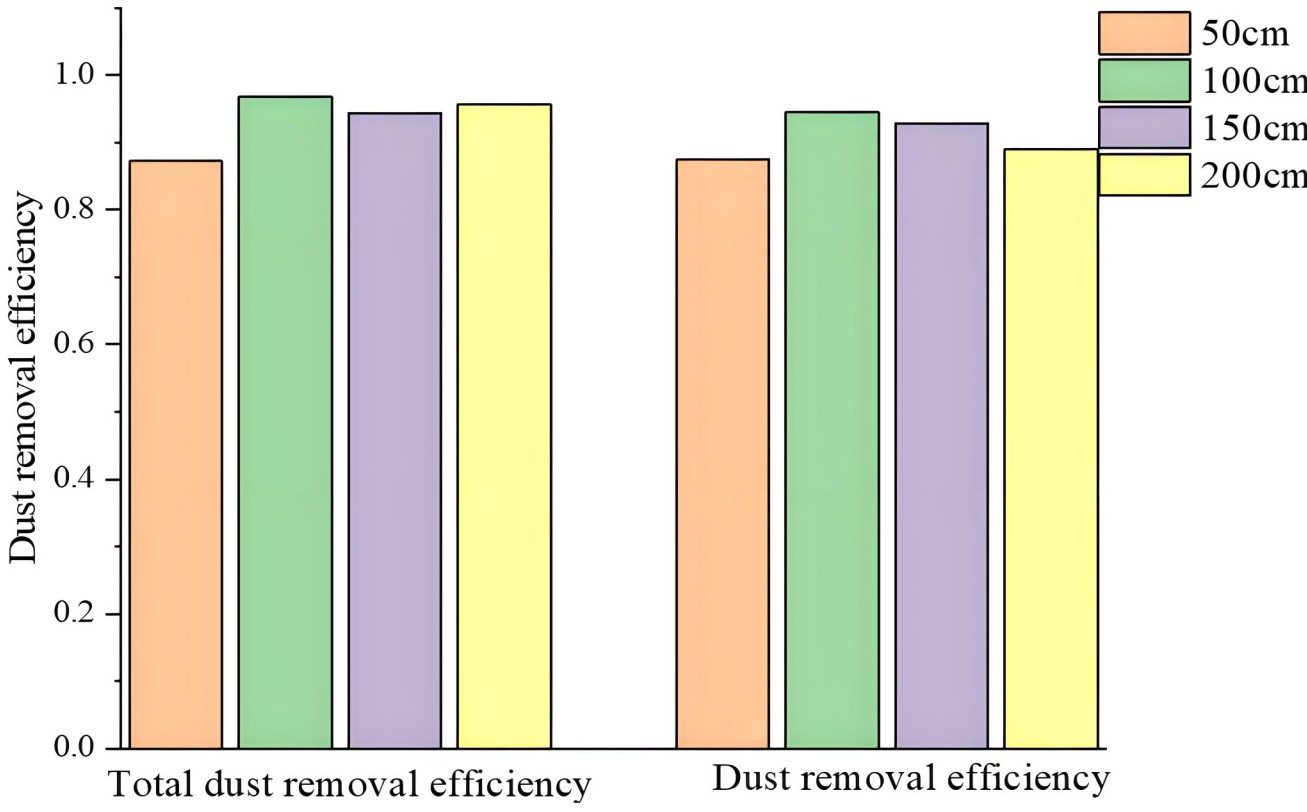

**Fig 43. Dust removal efficiency.**

dust diffusion of the excavation face and significantly improved the working environment of the excavation face.

### 5.3 Comparative analysis of dust concentration between simulation and test conditions

To verify the dust control situation obtained by the simulation and the test, we compared and analyzed changes in dust concentration.

As shown in Fig 44, the dust control effect was better under the installation of a double-wall attached annular air duct, and the simulation results were in good agreement with the experimental results. The established calculation model was correct and the data consistency was good.

### 6. Conclusions

In this study, we conducted a theoretical analysis and numerical simulation on the dust migration law of a combined excavation surface. To improve the dust collection and dust control efficiency of the combined excavation surface, we installed a double-wall attached-ring dust control air curtain. To achieve the best dust control effect, we optimized the inlet wind speed and the angle of the wall-attached blade. Following are the main conclusions obtained through this research:

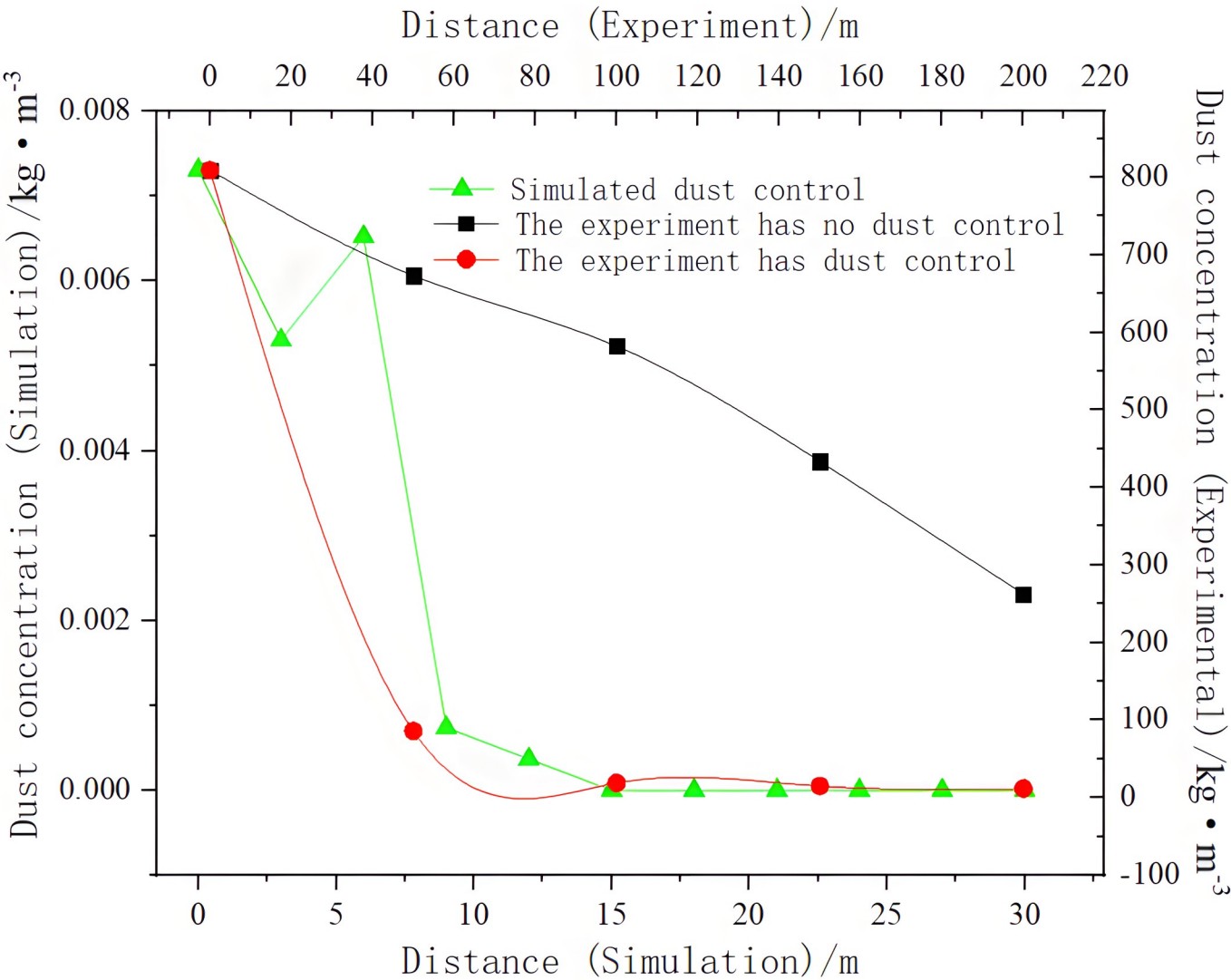

**Fig 44. Changes in dust concentration along the roadway.**

1. We designed a double-wall attached-ring dust control air curtain and established an overall dust control technical scheme and structure. Through model analysis of the dust control principle of the double-wall attached air curtain, we examined the wall attachment effect and the theoretical principle of volume suction flow. We concluded that the double-wall attached air curtain created a reasonable dust control flow field, thus achieving the purpose of effective dust control and dust removal in an integrated excavation space.

2. We built a physical model of the dust control test platform of the TBM and double-wall attached-ring air curtain using simulation software. After analysis, the double-wall attached-ring air curtain was able to form a stable circulating eddy current field, and most of the dust particles were effectively removed through the exhaust duct. Because of the wall attachment effect of the wall-attached blade, the formation area of the air curtain was

expanded, and the portion of dust at the front end of the driver was also effectively controlled.

3. We explored the influencing factors of the dust control effect of the ring air curtain and determined the optimal parameters. By using the control variable method, keeping the angle of the attached blade unchanged and changing the speed of the inlet, we established that the optimal initial speed of the dust control air curtain was 30 m/s. When the nozzle area and initial velocity remain unchanged, the angle of the blade attached to the wall was changed, and the optimal angle of the blade attached to the wall was 30˚.

4. According to the software simulation results and the specifications of the real underground integrated excavation face, we built a dust control test platform with double-wall attached steam cyclone curtain and obtained the wind speed attenuation law at different sections. The farther away from the dust control equipment, the smaller the wind speed, the faster the speed attenuation at the distance of 20 cm, and the slower the speed attenuation at the distance of 50 cm. We tested the total dust concentration and exhaled dust concentration. The total dust concentration decreased from 809.1 mg/m$^3$ to 11.8 mg/m$^3$, and the total dust removal efficiency reached 98.5%. The dust concentration decreased from 71.5 mg/m$^3$ to 1.7 mg/m$^3$, and the dust control efficiency reached 97.5%, which verified the effectiveness of double-wall attached-ring air curtain dust control technology.

## Supporting information

**S1 File.**
(PDF)

## Author Contributions

**Data curation:** Xuerong Pan.

**Formal analysis:** Jinyi Zhang.

**Resources:** Xuerong Pan.

**Software:** Jinyi Zhang.

**Supervision:** Niujun Jia.

**Writing – original draft:** Jingxue Yan.

**Writing – review & editing:** Baoshan Jia.

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
