## [Decision Letter · Decision Letter 0]

8 Nov 2023

PONE-D-23-31828Research on the dust-control technology of a double-wall attached-ring air curtain on an excavation facePLOS ONE

Dear Dr. Jia,

Thank you for submitting your manuscript to PLOS ONE. After careful consideration, we feel that it has merit but does not fully meet PLOS ONE’s publication criteria as it currently stands. Therefore, we invite you to submit a revised version of the manuscript that addresses the points raised during the review process.

We look forward to receiving your revised manuscript.

Kind regards,

Rajeev Singh

Academic Editor

PLOS ONE 

 [Natural Science Foundation of Liaoning Province) with grant number (2020-MS-304].  

[This work was partly supported by the Natural Science Foundation of Liaoning Province, No.2020−MS−304]

[Natural Science Foundation of Liaoning Province) with grant number (2020-MS-304]

6. PLOS requires an ORCID iD for the corresponding author in Editorial Manager on papers submitted after December 6th, 2016. Please ensure that you have an ORCID iD and that it is validated in Editorial Manager. To do this, go to ‘Update my Information’ (in the upper left-hand corner of the main menu), and click on the Fetch/Validate link next to the ORCID field. This will take you to the ORCID site and allow you to create a new iD or authenticate a pre-existing iD in Editorial Manager. Please see the following video for instructions on linking an ORCID iD to your Editorial Manager account: "" ext-link-type="uri" xlink:type="simple">https://www.youtube.com/watch?v=_xcclfuvtxQ""

7. We note that Figure(s) 1a, 1b, 2, 3, 4, 5, 6, 7a, 7b, 8, 9, 10, 11, 13, 14, 15, 16, 18, 19a, 19b and 20b in your submission contain copyrighted images. All PLOS content is published under the Creative Commons Attribution License (CC BY 4.0), which means that the manuscript, images, and Supporting Information files will be freely available online, and any third party is permitted to access, download, copy, distribute, and use these materials in any way, even commercially, with proper attribution. For more information, see our copyright guidelines: http://journals.plos.org/plosone/s/licenses-and-copyright.

a. You may seek permission from the original copyright holder of Figure(s) 1a, 1b, 2, 3, 4, 5, 6, 7a, 7b, 8, 9, 10, 11, 13, 14, 15, 16, 18, 19a, 19b and 20b to publish the content specifically under the CC BY 4.0 license. 

Reviewers' comments:

Reviewer's Responses to Questions

**Comments to the Author**

1. Is the manuscript technically sound, and do the data support the conclusions?

Reviewer #1: Yes

Reviewer #2: Yes

Reviewer #3: Yes

2. Has the statistical analysis been performed appropriately and rigorously? 

Reviewer #1: Yes

Reviewer #2: Yes

Reviewer #3: Yes

3. Have the authors made all data underlying the findings in their manuscript fully available?

Reviewer #1: Yes

Reviewer #2: No

Reviewer #3: Yes

4. Is the manuscript presented in an intelligible fashion and written in standard English?

Reviewer #1: Yes

Reviewer #2: Yes

Reviewer #3: Yes

5. Review Comments to the Author

Reviewer #1: I commend your choice of this research topic. The importance and influence of the research question have been well explained in the paper.

However, there are still some question.

(1) In the introduction part of the article, you can further clarify the research purpose and research method, so that readers can better understand your research significance and research framework.

(2) The statement "The simulation analysis showed that~~~could not meet the ideal environmental requirements" in line 140 is not clear, The difference in specific values is not explicitly stated in the preceding paragraph. It's better to put it in the paper.

(3) Since the experiment is a similar reduction experiment, but there are some problems in the ratio of length, width and height, may I ask how the ratio of numerical simulation and similar reduction experiment is adjusted?

(4) The Double-wall attachment-ring air curtain dust control scheme proposed in this paper is a very interesting idea.

(5) The text expression of the device in Fig.18 was changed to English

(6) What is the specific dust source in the article, and will different dust sources have different effects on the experimental results? Please add a statement.

Reviewer #2: The main work point of the manuscript is to simulate the dust removal effect of a double-armed circular air curtain under different working conditions. In my opinion, the manuscript has the following main problems:

1. In line 59, the author says "Scholars have not yet developed the application of air curtain technology and how it may be combined with mine dust treatment". "I do not agree with this point of view, because there are already many papers on the use of air curtain technology in mine dust treatment, and the author should focus on the shortcomings of the current results, as well as his own innovative points.

2. In terms of parameterization, does the air curtain have a different absolute value for the boundary condition velocity and is the velocity 15 m/s?

3. Poor quality of images

Reviewer #3: The title of the submitted manuscript is attractive and the novelty of the paper is interesting, but there are still some issues that need to be resolved:

1) Literature research should be more thorough. These articles can be used as reference to enhance the persuasiveness of the manuscript.

2) The error range of the results should be displayed.

3) How to determine the Angle of the wall attached blade in the simulation.

4) What is the maximum allowable concentration of respirable dust stipulated in "the Coal Mine Safety Regulations" in line 337? Please add.

5) "the dust concentration of each Y-Z section" in line 367 is inconsistent with the previous paragraph, please explain.

6) The novelty of the work must be addressed, and it is suggested that the novelty be highlighted at the end of the introduction.

7) There are grammatical errors in the text. The language of the full text needs to be further polished.

6. PLOS authors have the option to publish the peer review history of their article (what does this mean?). If published, this will include your full peer review and any attached files.

Reviewer #1: No

Reviewer #2: No

Reviewer #3: **Yes: **Jintuo Zhu

---

## [Author Response · Author response to Decision Letter 0]

11 Nov 2023

Reply to Editor:

(1) I have modified the format according to the requirements of PLOS ONE.

(2) This article is not subject to any restrictions and can be used by PIOS ONE at will.

(4) I have deleted the words related to funds in the acknowledgments. Please feel free to let me know if there is anything else that needs to be added in my fund statement.

(5) I have uploaded all my relevant data through the support information file.

(6) The ORCID ID has been added to the home page.

(7) It has been approved by the original author and her personal signature has been obtained. Of course, it is only in Chinese. If you need a handwritten signature in English, please contact me and I will contact the original copyright author.

(8) The "Reference list" has been checked again. If I still have omissions, please understand and contact me. I will make corrections as soon as possible.

Response to Reviewers Reviewer 1.

Reviewer #1

Dear external audit expert, hello. Thank you for your review of my article. I will make timely revisions according to your suggestions.

(1) I will add some references according to your requirements to make the article more convincing, thank you.

(2) I have added error bars to the image data

(3) The blade Angle selected according to the real situation is: 20°, 30°, 45°, and larger or smaller angles are not in line with the reality.

(4) It has been added in the first sentence after the title of Chapter 3.

(5) It has been modified according to your comments.

Ok, thank you for your suggestion.

Response to Reviewers Reviewer 2

Dear external audit expert, hello. Thank you for your review of my article.

(1) The external audit expert is sorry, I'm not saying "Scholars have not yet developed the application of air curtain technology and how it may be combined with mine dust treatment". My original meaning is that the innovation is not enough, but I did not check the translation carefully. I will revise this paragraph, thank you for your comments.

(2) Sorry, I did only do 15m/s Velocity Magnitude of Inlet Duct at first without the wind curtain, but after setting up the wind curtain, I set three parameters for Velocity Magnitude of Inlet Duct: 25m/s, 30m/s, 35m/s. Inlet and outlet were accidentally confused in the article. Thanks for your suggestion, I have modified it.

(3) The image quality was originally very good, but the journal required TIFF format, and the pixel adjustment on the website PACE, which came with the journal, resulted in insufficient clarity through multiple format compression and image compression. But I'll upload the artwork to the journal. Thank you.

Response to Reviewers Reviewer 3

Dear external audit expert, hello. Thank you for your review of my article. I will make timely revisions according to your suggestions.

(1) I will add some references according to your requirements to make the article more convincing, thank you.

(2) I have added error bars to the image data

(3) The blade Angle selected according to the real situation is: 20°, 30°, 45°, and larger or smaller angles are not in line with the reality.

(4) It has been added in the first sentence after the title of Chapter 3.

(5) It has been modified according to your comments.

Ok, thank you for your suggestion.

---

## [Editor Report · Decision Letter 1]

14 Nov 2023

Research on the dust-control technology of a double-wall attached-ring air curtain on an excavation face

PONE-D-23-31828R1

Dear Dr. Jia,

We’re pleased to inform you that your manuscript has been judged scientifically suitable for publication and will be formally accepted for publication once it meets all outstanding technical requirements.

Kind regards,

Rajeev Singh

Academic Editor

PLOS ONE
---

## [Editor Report · Acceptance letter]

20 Nov 2023

PONE-D-23-31828R1 

Research on the dust-control technology of a double-wall attached-ring air curtain on an excavation face 

Dear Dr. Jia:

I'm pleased to inform you that your manuscript has been deemed suitable for publication in PLOS ONE. Congratulations! Your manuscript is now with our production department. 

Kind regards, 

on behalf of

Dr. Rajeev Singh 

Academic Editor

PLOS ONE